# PM_2.5_ Concentration Exposure over the Belt and Road Region from 2000 to 2020

**DOI:** 10.3390/ijerph19052852

**Published:** 2022-03-01

**Authors:** Shenxin Li, Sedra Shafi, Bin Zou, Jing Liu, Ying Xiong, Bilal Muhammad

**Affiliations:** 1School of Geosciences and Info-Physics, Central South University, Changsha 410083, China; sedrashafi@csu.edu.cn (S.S.); 210010@csu.edu.cn (B.Z.); 195011041@csu.edu.cn (J.L.); 2School of Architecture, Changsha University of Science & Technology, Changsha 410083, China; csustxy@126.com; 3School of Marine Sciences, Nanjing University of Information Science and Technology, Nanjing 210044, China; muhammad.bilal@connect.polyu.hk

**Keywords:** remote sensing, PM_2.5_ pollution, population weighted exposure, the Belt and Road Region

## Abstract

Ambient fine particulate matter (PM_2.5_) can cause respiratory and heart diseases, which have a great negative impact on human health. While, as a fast-developing region, the Belt and Road (B&R) has suffered serious air pollution, more detailed information has not been revealed. This study aims to investigate the evolutionary relationships between PM_2.5_ air pollution and its population-weighted exposure level (PWEL) over the B&R based on satellite-derived PM_2.5_ concentration and to identify the key regions for exposure control in the future. For this, the study focused on the B&R region, covering 51 countries, ranging from developed to least developed levels, extensively evaluated the different development levels of PM_2.5_ concentrations during 2000–2020 by spatial-temporal trend analysis and bivariate spatial correlation, then identified the key regions with high risk under different levels of Air Quality Guidelines (AQG). Results show that the overall PM_2.5_ and PWEL of PM_2.5_ concentration remained stable. Developing countries presented with the heaviest PM_2.5_ pollution and highest value of PWEL of PM_2.5_ concentration, while least developed countries presented with the fastest increase of both PM_2.5_ and PWEL of PM_2.5_ concentration. Areas with a high level and rapid increase PWEL of PM_2.5_ concentration were mainly located in the developing countries of India, Bangladesh, Nepal, and Pakistan, the developed country of Saudi Arabia, and least developed countries of Yemen and Myanmar. The key regions at high risk were mainly on the Indian Peninsula, Arabian Peninsula, coastal area of the Persian Gulf, northwestern China, and North China Plain. The findings of this research would be beneficial to identify the spatial distributions of PM_2.5_ concentration exposure and offer suggestions for formulating policies for the prevention and control PM_2.5_ air pollution at regional scale by the governments.

## 1. Introduction

It is reported that 7.3 million premature deaths in each year are related to poor air quality conditions [1] all over the world. Among aerosol pollutants, fine particulate matter (PM_2.5_) is the major pollutant that has a negative impact on heart, lung, and even psychological disorders [2,3,4,5,6]. Due to the rapid growth in urbanization and industrialization over past two decades, the serious problem of air pollution becomes increasingly prominent in the Belt and Road (B&R) region (e.g., India [7,8], China [9,10], Pakistan [11,12]). This situation is particularly outstanding in urban areas [13,14,15,16,17]. Although the air pollution in the B&R region was generally serious, from the local perspective, distributions of air pollution were not consistent over the whole region, and significant differences were also shown across different countries or cities [18]. In this case, it is necessary to analyze the spatial and temporal characteristics of air pollution in the B&R region and to identify the areas with heavy pollution. Moreover, the disparities of air quality among countries greatly depend on the development levels of each country. Previous studies based on monitoring stations and remote sensing data disclosed that poor air quality areas were mainly concentrated in developing and least developed countries [19,20,21,22]. Nevertheless, in the process of the B&R initiative, there will be a new stage of accelerated urbanization in countries across the B&R region. Undoubtedly, these countries will face greater environmental exposure pressure. Thus, in order to avoid environmental injustice and provide guidance for environmental governance, it is important to conduct a comprehensive assessment of the air pollution level over the B&R region.

Therefore, this study aims to assess the 51 countries grouped by different development level over the B&R region and comprehensively estimated the spatiotemporal dynamic changes of PM_2.5_ concentrations and its PWEL based on satellite-based dataset from 2000 to 2020. The specific objectives of this study are as follows (1) analyze the spatiotemporal characteristics of PM_2.5_ concentrations in the B&R region among the developed, developing, and least developed countries based on a 0.01° × 0.01° satellite-based PM_2.5_ concentration dataset, (2) estimate and compare the changes of PWEL of PM_2.5_ concentration among different development groups, countries, and cities in the B&R region, and (3) examine the cluster trend of PM_2.5_ concentration exposure and identify the risk regions in different countries. 

The results of this study would help in understanding the spatiotemporal characteristics of long-term PM_2.5_ pollution exposure over the B&R region. The study specially focused on comparing the differences of PM_2.5_ pollution exposure among different countries according to development level. In this case, it would provide a perspective for least developed and developing countries to draw lessons from the experience of PM_2.5_ concentration governance in the developed countries, as well as to be guided by empirical evidence in the prevention and control of PM_2.5_ pollution globally. Besides, under the B&R initiative, the results of PM_2.5_ pollution exposure could contribute to providing a reference for the development and optimization of regional industrialization and urbanization, and further to decrease the air pollutant burden to help these countries achieve the sustainability objectives in the B&R region.

## 2. Data and Methods

### 2.1. Study Area

The B&R region is selected as the study area because it is an important region with great potential for global future development and sustainable economic growth. Based on gross domestic product (GDP, collected from DataBank of the Word Bank, https://www.worldbank.org/ (accessed on 10 October 2021)), the selected 51 countries are firstly divided into four income levels (low income, upper middle income, lower middle income, and high income), then classified into three different development groups, including 9 developed (high income), 33 developing (upper and lower middle income), and 9 least developed (low income) countries. The location and different development level of countries in the B&R region are shown in Figure 1.

### 2.2. Data

#### 2.2.1. Satellite-Based PM_2.5_ Concentration Data

PM_2.5_ concentration data with a resolution of 0.01° × 0.01° covering the period from 2000 to 2020 were obtained from the Atmospheric Composition Analysis Group at Dalhousie University (https://sites.wustl.edu/acag/datasets/surface-pm2-5/ (accessed on 1 December 2021)). This study selected the annual data version of Global/Regional Estimates (V5.GL.02). The data are calculated by the geographically weighted regression model [23,24] using the satellite Aerosol Optical Depth (AOD) products from MODIS, MISR, and SeaWiFS sensors as well as data based on the Goddard Earth Observing System atmospheric chemical transport model. The original datasets were provided in NetCDF [.nc] format data. We converted them to “Geotiff” format and resampled to match the resolution of population data (1 × 1 km).

#### 2.2.2. Population Data

The population density gridded data with a resolution of 1 × 1 km from 2000 to 2020 were obtained from the WorldPop datasets (https://www.worldpop.org/ (accessed on 1 December 2021)). The production of WorldPop spatial datasets principally integrated data and methods of contemporary census data, settlements mapping, bottom-up population mapping, intra-urban population mapping, and so on. These data consist of human population distribution, which can be used to calculate the PWEL of PM_2.5_ concentrations.

### 2.3. Methods

The technical flowchart of this study is shown in Figure 2. There were three main steps. The first step was to analyze the spatiotemporal characteristics of PM_2.5_ concentration in the B&R region based on satellite-based PM_2.5_ concentration data and methods of statistical analysis were used. The second step was PWEL of PM_2.5_ concentration calculation by combining PM_2.5_ concentration and population data. The third step was focused on identifying the key regions for exposure risk control according to World Health Organization (WHO) Air Quality Guidelines (AQG).

#### 2.3.1. Calculation of Population-Weighted Exposure Level of PM_2.5_ Concentration

The combined effect of PM_2.5_ concentration and population density is based on exposure risk. This study uses the PM_2.5_ concentration PWEL to estimate the exposure risk of PM_2.5_ concentrations. The PM_2.5_ concentration PWEL of the given grid *i* is determined through the exposure equation as follows:(1)$${\mathrm{PWEL}}_{m}=\frac{\sum \left(P_{i}\times Y_{i}\right)}{\sum P_{i}}$$
where $P_{i}$ is the population in the grid i and $Y_{i}$ is its average PM_2.5_ concentration.

#### 2.3.2. Temporal Trend (Slope) Analysis

To explore the trends in a great number of concentration data from 51 countries over the B&R region from 2000 to 2020, a linear slope model is utilized [25].
(2)$$slope_{m}=\frac{n\times \sum_{i=1}^{n}\left(i\times Y_{i}\right)-\frac{1}{n}\sum_{i=1}^{n}i\sum_{i=1}^{n}Y_{i}}{\sum_{i=1}^{n}i^{2}-\frac{1}{n}{\left(\sum_{i=1}^{n}i\right)}^{2}}$$
where *Y_i_* signifies the attribute of PM_2.5_ concentration in a year *i*, *n* represents the time span, and *i* represents the time unit (yr). A positive value of slope (Slope > 0) implies that the data presents a rising trend over time, whereas a negative slope (Slope < 0) implies it has a decreasing trend over time. If the slope approaches zero, then it is without any significant pattern. 

#### 2.3.3. Bivariate Spatial Correlation

To examine the spatial association of bivariate observations, bivariate spatial correlation can be used, which is an expansion of the spatial correlation analysis. This study utilizes the bivariate local indicators of spatial association (LISA) or Anselin’s LISA to distinguish the local spatial association type of the various patterns of PM_2.5_ concentration and urban count. The bivariate LISA is defined as: (3)$$I_{kl}^{i}=Z_{k}^{i}\overset{n}{\underset{j=l}{\sum }}W_{ij}Z_{l}^{i}$$
where, $W_{ij}$ is the spatial weight matrix, $Z_{k}^{i}=\left[x_{k}^{i}-\overset{¯}{x_{k}}\right]/\sigma_{k}$, $Z_{k}^{j}=\left[x_{l}^{j}-\overset{¯}{x_{1}}\right]/\sigma_{l}$, $\;x_{k}^{i}$ is the observation *k* (the slope of PM_2.5_ concentration) at location *i*, $x_{l}^{j}$ is the observation l (PWEL) at location *j*, and $\sigma_{k}$ and $\sigma_{l}$ are the variance of $x_{k}$ and $x_{l}$, respectively. 

The results of bivariate LISA can be picturized by utilizing the Moran’s I scatter plot [26] which are categorized by the panel of spatial correlation into four sets (four quadrants divided through the vertical and the horizontal axis). The QI and QIII quadrants imply that the bivariate variables in those spatial units have a positive spatial correlation, or (high-high and low-low) spatial clusters. The QII and QIV quadrants depict a negative spatial correlation, or (high-low and low-high) spatial outliers [27]. The spatial clusters or spatial outliers are the values of the bivariate variables in some spatial units where there are significant positive or negative spatial correlation. GeoDa programming was executed to lead the bivariate LISA analysis in this paper.

#### 2.3.4. Key Regions Identified Based on WHO Air Quality Guidelines

The latest AQG published by the WHO in September 2021 set the AQG level of PM_2.5_ concentration to 5 μg/m^3^. Meanwhile, WHO also provides four interim targets (IT, shown in Table 1) as “incremental steps in progressive reduction of air pollution”. The interim targets should be regarded as steps towards ultimately achieving AQG levels in the future, rather than as end targets. According to above guidelines, the study calculated the difference between the PWEL in 2020 and that of corresponding WHO IT. If the value of difference is positive, then regard the region with unacceptable exposure risk. Otherwise, regard them as non-risk regions.

## 3. Results

### 3.1. Spatial-Temporal Variations of PM_2.5_ Concentration

Figure 3a shows the temporal trend of regional PM_2.5_ concentrations. Generally, annual mean PM_2.5_ concentrations over the whole B&R region remained relatively stable (from 20.62 μg/m^3^ to 21.26 μg/m^3^) during 2000 and 2020, ranging between 20.62 μg/m^3^ (in 2000) and 24.96 μg/m^3^ (in 2012). For different development groups, developing countries experienced the heaviest PM_2.5_ pollution, with an average PM_2.5_ concentration value of 31.98 μg/m^3^, an rate increase of 1.81%, ranging between 28.71 μg/m^3^ (in 2000) and 34.28 μg/m^3^ (in 2013). Least developed countries presented with an average value of 28.71 μg/m^3^ and the highest increased rate of 11.43%. PM_2.5_ concentrations in developed countries were at a relatively low and stable level with the average concentration and change rate of 13.41 μg/m^3^ and 3.68%.

Temporal trends of distribution for city-level concentrations in each development group are shown in Figure 3b–d. The concentrated interval of city-level concentrations in developed countries was the smallest (10~90% interval was 10–25 μg/m^3^) and kept relatively stable (Figure 3b). The city-level concentrations in least developed countries showed obvious fluctuation with time. It concentrated on 9–44 μg/m^3^ in the year of 2000, varied to concentrate on 12–52 μg/m^3^ in 2012, and finally changed to the interval of 10–45 μg/m^3^ in 2020 (Figure 3d). Developing countries presented with the largest interval of city-level concentrations, which concentrated on 12–56 μg/m^3^ during the whole study period (Figure 3c).

Figure 4 shows the spatial distributions of PM_2.5_ concentrations in the B&R region. Figure 4a shows the distributions of overall average PM_2.5_ concentrations from 2000 to 2020. There were 12 countries with PM_2.5_ concentrations exceeding 35 μg/m^3^, including nine developing, two least developed, and one developed country. The heavy PM_2.5_ pollution mainly occurred in developing countries, such as Bangladesh (57.17 μg/m^3^), Kuwait (53.87 μg/m^3^), Pakistan (51.78 μg/m^3^), Oman (51.69 μg/m^3^), and Qatar (51.66 μg/m^3^). Meanwhile, cities with PM_2.5_ concentrations exceeding the standard accounted for 13.27%. Among them, 321 cities belonged to the group of least developed countries (258 in Afghanistan, 63 in Yemen) and six cities belonged the developed country of Saudi Arabia. Cities with high PM_2.5_ concentrations were concentrated in the developing countries of India (44 cities with PM_2.5_ concentrations over 100 μg/m^3^ and 86 cities with PM_2.5_ concentrations between 75 and 100 μg/m^3^), China (23 cities with PM_2.5_ concentrations between 75 and 100 μg/m^3^), Indonesia (2 cities with PM_2.5_ concentrations between 75 and 100 μg/m^3^), and Pakistan (two cities with PM_2.5_ concentrations between 75 and 100 μg/m^3^). On the contrary, there were eight countries and over 4000 cities with PM_2.5_ concentrations below 15 μg/m^3^. Developed countries, such as Russia, Estonia, Philippines, and least developed countries of Brunei and Mongolia remained at a relatively light pollution level. 

Temporal variations of countries and cities were also showed different tendencies (Figure 4b–d). There were seven countries and 824 cities that maintained an annual mean PM_2.5_ concentration that exceeded 35 μg/m^3^ during the whole study period. Indonesia was the country with largest growth rate (from 12.85 μg/m^3^ to 27.23 μg/m^3^, increased 112%), followed by Brunei (75.97%), Jordan (70.17%), Saudi Arabia (46.04%), and Bangladesh (42.47%). PM_2.5_ concentrations of Nepal (Developing country) and Saudi Arabia (Developed country) varied from under 35 μg/m^3^ in 2000 to higher than the standard in 2020. Over 800 cities in the B&R region remained the annual mean PM_2.5_ concentrations higher than the standard and most of them located in developing countries such as India (336 cities), China (159 cities), and Bangladesh (60 cities). The top 65 cities with the largest increasing rate of PM_2.5_ concentration were in Indonesia, and the city of Pulang Pisau takes the first place (416%). PM_2.5_ concentrations of 626 cities achieved the standard in 2000 but exceeded it over time. Among them, cities in developing countries (Southeast Asia and Middle East countries) accounted for 70.61%, and 27.96% belong to least developed countries (175 cities in total, 107 in Afghanistan and 68 in Yemen). Meanwhile, 1.43% were in developed countries (nine cities in Saudi Arabia).

On the contrary, there were 24 countries with a declining trend, including 13 developing countries, eight developed countries, and three least developed countries. Developing countries of Macedonia (36.98%), Slovenia (32.15%), and Albania (30.54%) were the top three countries for a significant decrease in PM_2.5_ concentration, followed by the developed countries of Estonia (29.71%), Romania (27.31%), and Poland (26.19%). Besides, 43.62% of cities presented with a decreasing trend of PM_2.5_ concentrations. Most cities in Russia, Philippines, China, and Slovenia presented with a decreasing rate greater than 50%. Significantly, in only one country (Uzbekistan) and 279 cities, the PM_2.5_ concentrations dropped from exceeding to achieving the standard. Philippines (19.00%), Egypt (17.56%), China (14.70%), Afghanistan (10.04%), and India (9.32%) accounted for a relatively large proportion among these cities.

### 3.2. Spatial-Temporal Variations of PWEL of PM_2.5_ Concentration

Figure 5 shows the temporal variations of PWEL of PM_2.5_ concentration and the comparison between PM_2.5_ and PWEL. The overall PWEL of PM_2.5_ concentration varied from 23.08 μg/m^3^ to 22.32 μg/m^3^ during 2000–2020. The peak value appeared in the year of 2011 (25.70 μg/m^3^). Similar to the variation trend of PM_2.5_ concentrations, the PWEL of PM_2.5_ concentration for developed countries showed a decreasing trend (from 17.40 to 14.66 μg/m^3^, decreased 15.72%). PWEL of PM_2.5_ concentration for developing and least developed countries showed a similar trend with an increase of 5.78% and 5.87% respectively. Figure 5c,d show that the temporal variations of interval of PWEL for each developed group were similar with those of PM_2.5_ concentrations. Figure 5e shows the comparison between PM_2.5_ and PWEL at city level. Totally, 54.33% cities presented with PWEL higher than PM_2.5_, among them were cities with PWEL 0.1 and 0.2 times higher than PM_2.5_ accounting for 9.53% and 3.05%, respectively. For different developed groups, the cities with PWEL higher than PM_2.5_ accounted for 49.58%, 61.51%, and 52.345% in developed, developing, and least developed counties, respectively.

Figure 6 shows the spatial distribution of city level PWEL PM_2.5_ concentration in 2000, 2010, and 2020. The high level PWEL of PM_2.5_ concentrations were mainly in developing countries, such as India, Bangladesh, Nepal, Qatar, China, and Pakistan. Among these countries, most cities with high PWEL of PM_2.5_ concentrations were in India and China. Besides, individual cities of developed countries also presented with high PWEL of PM_2.5_ concentrations. Developed and least developed countries presented mainly with a relatively low PWEL of PM_2.5_ concentrations, such as Estonia, Mongolia, Brunei, and Philippines. For cities, about 70% cities of PWEL of PM_2.5_ concentrations were in developed countries and these proportions of developing and least developed countries were 22% and 8%.

There were 27 countries with an increasing trend of PWEL of PM_2.5_ and Brunei, Myanmar, and Nepal were the top three countries with growth rates of 63.97%, 50.43%, and 45.74%, respectively. Among them, developing countries accounted for 77.78%, and proportions of least developed and developed countries were 18.52% and 3.7% respectively. On the other hand, among the 24 countries which presented with declining PWEL of PM_2.5_ concentrations, Philippines was the countries with largest decrease of PWEL of PM_2.5_ (declined 37.51%), followed by Macedonia (declined 30.73%), Estonia (declined 29.59%), and Slovenia (declined 28.96%). At the city scale, the proportions of cities with increased and decreased trends were 43.23% and 56.77%. Among cities with an increased trend, 256 cities with the PWEL of PM_2.5_ concentrations doubled over the study period. Developing (e.g., Indonesia, Malaysia, India), developed (i.e., Philippines, Russia), and least developed (i.e., Brunei, Mongolia) countries accounted for 78.52%, 19.53%, and 1.95%, respectively. Compared with increased trend cities, the number of cities where PWEL of PM_2.5_ concentrations descended over 50% was 391 (92.58% were developed countries and 7.42% were developing countries).

### 3.3. The Evolutionary Relationships Based on PWEL of PM_2.5_ Concentration

Based on city-level PWEL of PM_2.5_ concentration, the study calculated the slope of PWEL during 2000 to 2020. Figure 7a shows the total distribution of PWEL slopes for the whole region. The slopes concentrated between −0.75 and 1.25 μg/m^3^/year. For each developed group, the slope interval of developed countries was −0.25–0.5 μg/m^3^/year, which was the smallest (Figure 7b), and the value of slope was mainly distributed in less than 0. Cities with negative slopes of developed countries were mainly located in Europe, especially in eastern areas, such as cities in Poland and Romania. Saudi Arabia was the developed county to have most cities with positive slopes (Figure 7e). The slope interval of developing countries was similar with the whole region, in the range of −0.75–1.25 μg/m^3^/year. Cities with a slope larger than 0 accounted for the majority, and they mainly belonged to India, Bangladesh, Nepal, Pakistan, and Oman. China, Kazakhstan, Kyrgyzstan, and Turkmenistan were the developing countries housing a majority of cities with negative PWEL slopes. Least developed countries presented with the interval of −0.75–0.75 μg/m^3^/year (Figure 7d) and cities with a negative slope were mainly in Mongolia and Armenia. Cities with a positive slope were in Yemen and Myanmar (Figure 7e).

Figure 8 shows the spatial cluster results of PWEL slope. For the whole region, 18% of cities presented with high-high cluster and 32% of cities presented with low-low cluster of PWEL slope during 2000–2020. Cities of developed countries mainly showed low-low cluster (accounting for 53%), and they were mainly located in Europe. Meanwhile, there were still about 1% of cities with high-high cluster in developed counties, and those cities mainly in Saudi Arabia and central Russia. The proportion of cities with high-high cluster in least developed countries was the largest (42%), and they belonged to Yemen, Myanmar, and eastern, northwestern, and southwestern Afghanistan. Cities with high-high cluster and low-low cluster in developing countries accounted for 32% and 14%, respectively. Among them, high-high cluster cities were located in India, Bangladesh, United Arab Emirates, Oman, and eastern Pakistan, northwestern China, eastern Uzbekistan, central Iran, and so on. Low-low cluster cities were concentrated in the north China Plain and southeastern China. Besides. High-low and low-high outlier cities accounted for small proportions in all development levels. There were less than 30 cities with high-low outlier and most of them located in northeast China. There were more cities with low-high outliers than that with high-low outliers. These cities were mainly in China (i.e., western Sichuan Basin and Qinghai-Tibet Plateau), western Pakistan, eastern Iran, Kyrgyzstan, western and eastern Turkmenistan, and southern Iraq.

### 3.4. Key Regions under Different Target of AQG

Figure 9 shows the key regions for PM_2.5_ exposure control under different IT values AQG. Generally, the extent of key regions varied more as the target of AQG became stricter. For the IT-1, key regions with high risk were located in developing countries, such as northwestern China, northern Indian Peninsula, and the southern Arabian Peninsula, which mostly belong to the developed country of Saudi Arabia. The high-risk regions for IT-2 expanded to North China Plain and the coastal area of Persian Gulf. For IT-3 and IT-4, key regions with high risk covered the areas of China, South Asia, Western Asia, and southern areas of Central Asia. According to the latest AQG level (5 μg/m^3^), nearly all the B&R regions presented with exceeding risks except Qinghai-Tibet Plateau in China and northern Russia (Figure 9e). Regions exceeding 20μg/m^3^ covered whole counties in the middle latitudes, while high and low latitude counties mainly presented with the exceedance interval between 5 and 15μg/m^3^.

## 4. Discussion and Conclusions

This study analyzed the spatiotemporal variations of PM_2.5_ air pollution and exposure over the B&R region from the perspective of PM_2.5_ concentration, PWEL of PM_2.5_ concentration, slope, and its cluster for PWEL of PM_2.5_ concentration and identified the key regions for exposure control. The long time series analysis results of this study completely revealed the spatiotemporal characteristic of PM_2.5_ exposures over the B&R region for the first time. These results could contribute to understand the spatiotemporal changes in the population exposure of PM_2.5_ concentrations in least developed, developing, and developed countries over the B&R region, which may benefit the formulation of policies for preventing and controlling regional scale PM_2.5_ pollution by the governments.

The results show that spatiotemporal distributions of PM_2.5_ air pollution deeply depend on development level over the B&R region. Developing countries, such as India, China, Pakistan, and Bangladesh, have experienced heavy PM_2.5_ pollution due to combustion-induced emissions from multiple sources, including household solid fuel use [28], coal-fired power plants, agricultural and open burning, as well as industrial and transportation-related sources [29,30,31]. During the study period, the PM_2.5_ concentration and exposure in China showed a significantly decrease, and these effects are due to the Chinese government’s long-term control policies, such as reducing emissions, use of cleaner energy, and industrial structure upgrading [32,33]. On the contrary, other south and southeast Asian countries were still suffering heavy pollution (i.e., India, Pakistan, and Bangladesh). These countries have issued a series of clean air policies and low-carbon energy targets. However, some barriers such as enforcement remain, and the implementation effect of these policies has been seriously affected [34]. The Arabian Peninsula (including the developed county Saudi Arabia, developing countries of Oman and United Arab Emirates, and least developed country Yemen) was another key region of high level PM_2.5_ concentration during the whole study period. The air quality of this region was affected by the transportation of African dust, fossil fuel industrial development, and climate change [35,36,37].

This study also compared the results between PM_2.5_ concentrations and PWEL of PM_2.5_ concentration. The inconsistency of the spatial distribution of PM_2.5_ concentrations and population density leads the differences between PM_2.5_ concentrations and PWEL of PM_2.5_ concentration. Totally, the PWELs of PM_2.5_ concentration were higher than PM_2.5_ concentrations in each year and different developed groups. This result indicates that high density population areas presented mainly with heavy pollution status [38,39,40]. For example, developing and least developed countries, e.g., China, Bangladesh, India, and Pakistan, faced high exposure risks of PM_2.5_ concentration during the whole study period due to the high density of their populations [41]. China and India both presented with serious PM_2.5_ concentrations and high population density and finally with high PWELs of PM_2.5_ concentration. Compared with developing and least developed counties, some developed countries, such as Brunei, Estonia, Slovenia, Poland, and Czech, experienced low exposure risk to PM_2.5_ [42] because of the low pollution concentration level. For example, the population density of Slovakia and Slovenia was relatively high (112/km^2^, 100/km^2^), but due to the low level of PM_2.5_ concentrations (e.g., 21.8 μg/m^3^ and 14.5 μg/m^3^), the PM_2.5_ concentration PWELs remained at low value levels (20.8 μg/m^3^ and 16.8μg/m^3^).

The study identified the key regions for PM_2.5_ exposure control under different IT of AQG at the city level. Significant variations of key regions were shown under different IT of AQG. In general, it is very difficult to achieve the final goal of AQG level in a short term. Considering the concept of IT as “incremental steps in progressive reduction of air pollution”, the government can formulate the air pollution control target by region and level based on results of this study. For example, most developed countries can set the target of IT-4 (10 μg/m^3^), while some cities of developing and least developed countries could set the target at IT-1 (35 μg/m^3^) or IT-2 (25 μg/m^3^). Besides, the key regions with high risk show obvious regional characteristics. The joint prevention and control mechanism has been proven to be an effective method for air pollution control in China [43,44,45]. In this case, a mechanism of joint prevention and control between countries needs to be established. Moreover, effective measures of air pollution control also need to be taken. Measures such as changing the economic and industrial structure, reducing industry and traffic emissions, or using clean production technologies could be applied in those high exposure risk regions [46,47,48,49,50].

This study also has a few issues and limitations. The first is the uncertainty of PM_2.5_ concentration data. The raw data for PM_2.5_ concentrations are estimated based on satellite AOD and chemical transport model. This data showed different uncertainty in different regions [23]. Thus, the calculated PWEL of PM_2.5_ concentration might be less precise due to the inherent data quality in some regions, which is usually more accurate in developed countries but less precise in some developing and least developed countries [23,51]. To enhance accuracy and reduce uncertainty in the modelled data, countries should continue to densify ground measurement networks, which will be investigated in the future. 

## Figures and Tables

**Figure 1 ijerph-19-02852-f001:**
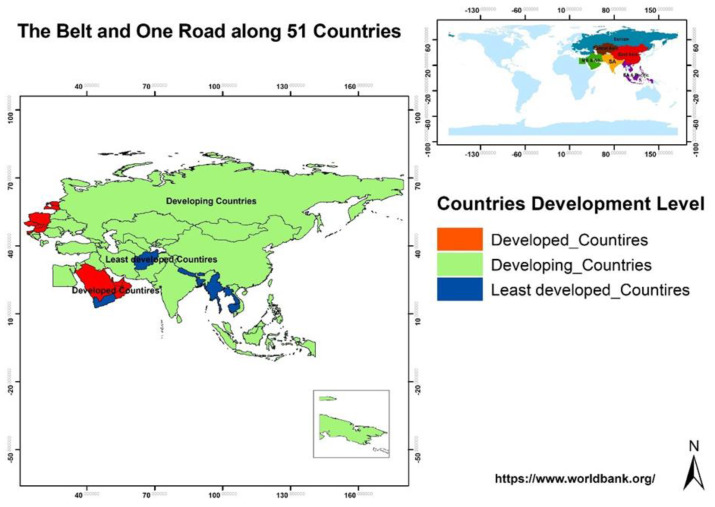
Location of study area.

**Figure 2 ijerph-19-02852-f002:**
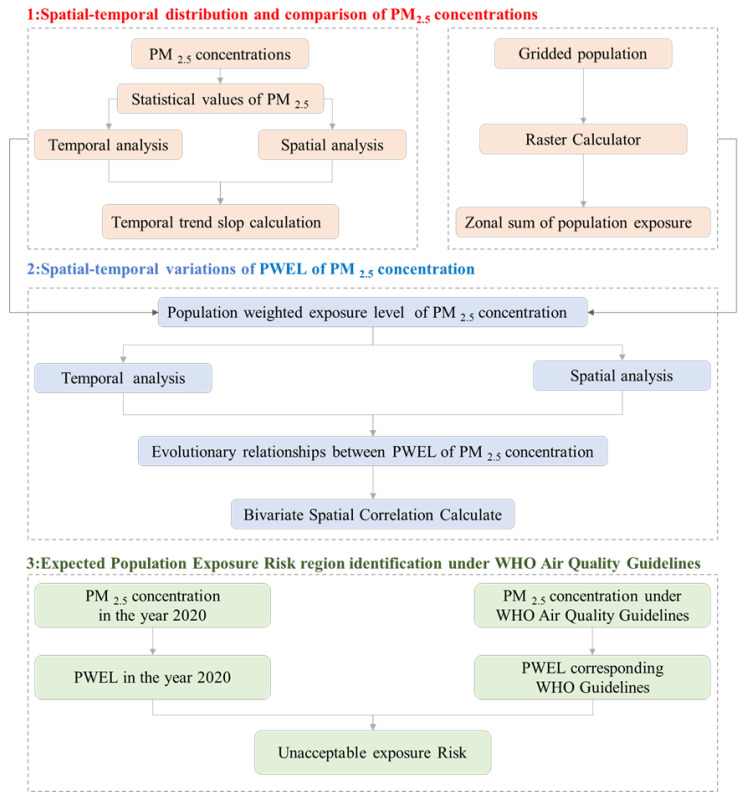
The technical flowchart of this study.

**Figure 3 ijerph-19-02852-f003:**
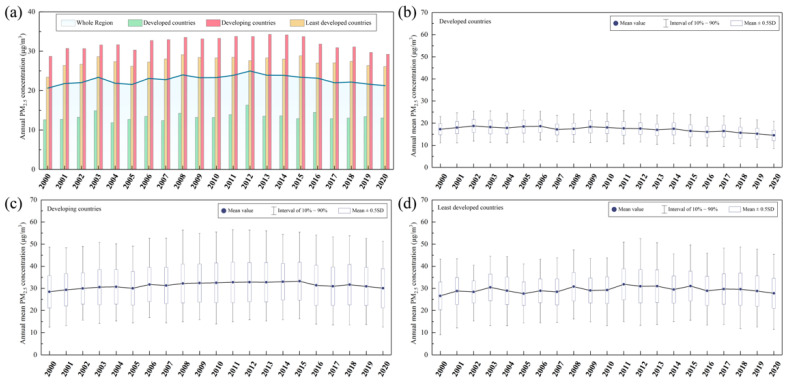
The temporal trends of (**a**) annual mean PM_2.5_ concentrations in the whole region, distribution of city-level PM_2.5_ concentrations in (**b**) developed countries, (**c**) developing countries, (**d**) least developed countries.

**Figure 4 ijerph-19-02852-f004:**
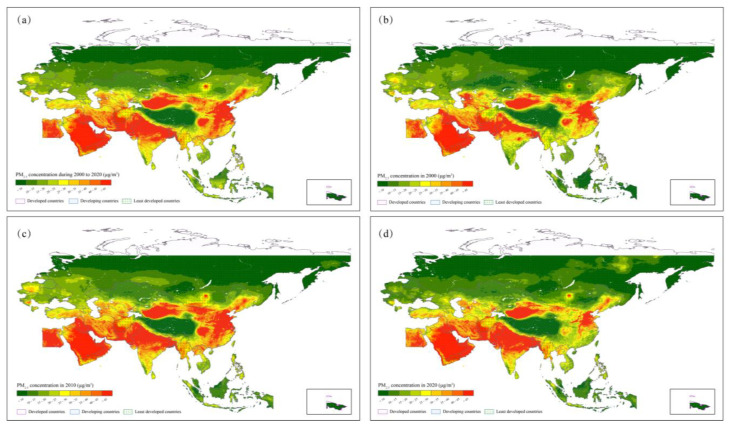
Spatial distribution of annual mean PM_2.5_ concentrations in B&R region in the year of (**a**) 2000 to 2020, (**b**) 2000, (**c**) 2010, (**d**) 2020.

**Figure 5 ijerph-19-02852-f005:**
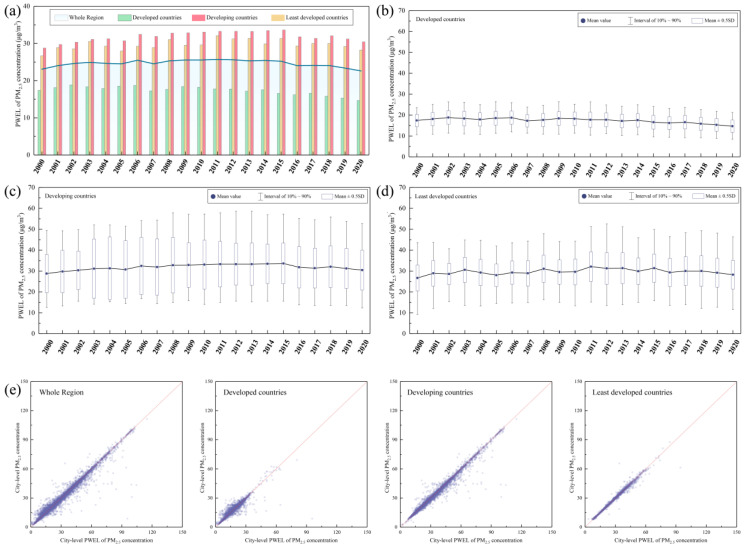
The temporal trends of (**a**) annual mean PWEL of PM_2.5_ concentration in the whole region, distribution of city-level PWEL of PM_2.5_ concentration in (**b**) developed countries, (**c**) developing countries, (**d**) least developed countries. (**e**) comparison between PM_2.5_ concentration and PWEL of PM_2.5_ concentration at city level.

**Figure 6 ijerph-19-02852-f006:**
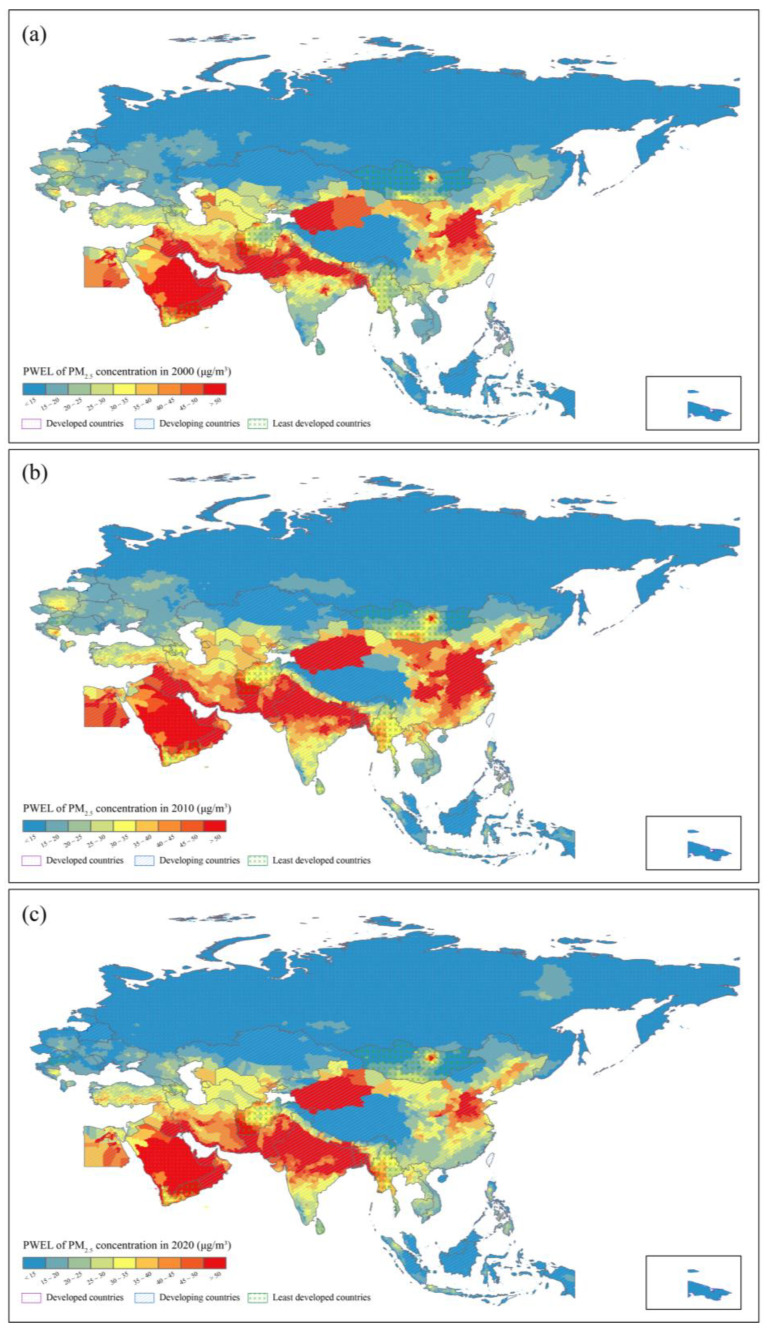
Spatial distributions of PWEL of PM_2.5_ concentration in (**a**) 2000, (**b**) 2010, (**c**) 2020 in B&R region at city-level.

**Figure 7 ijerph-19-02852-f007:**
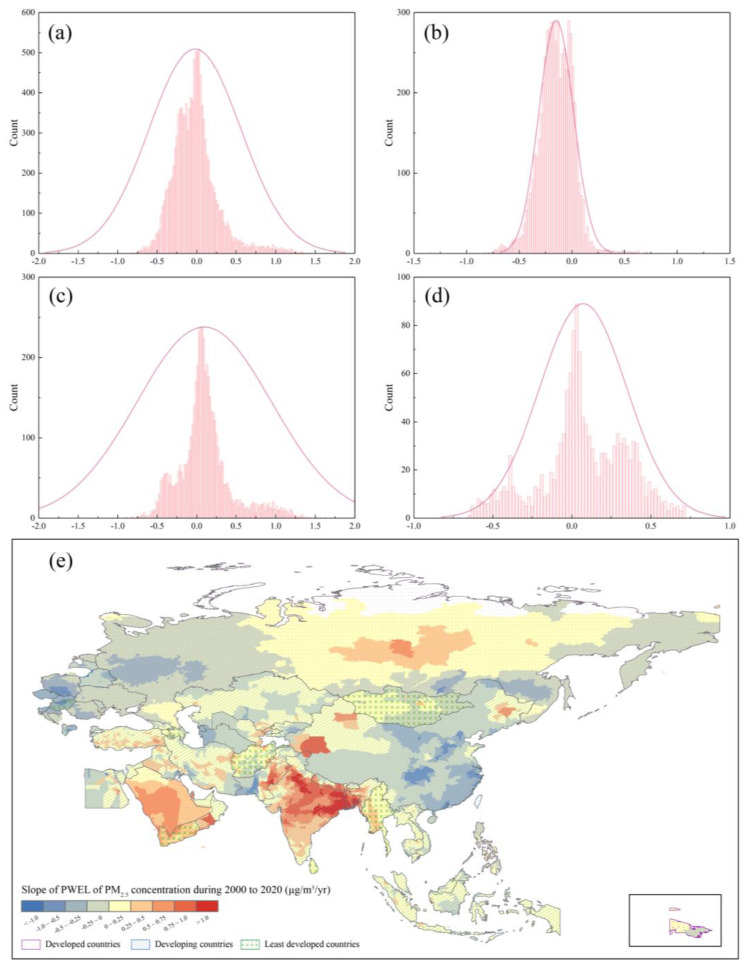
(**a**) the distribution of slope of PWEL of PM_2.5_ concentration for the whole B&R region, the distribution of slope of PWEL of PM_2.5_ concentration for (**b**) developed countries, (**c**) developing countries, (**d**) least developed countries, (**e**) the spatial distribution of slope of PWEL of PM_2.5_ concentration in B&R region from 2000 to 2020.

**Figure 8 ijerph-19-02852-f008:**
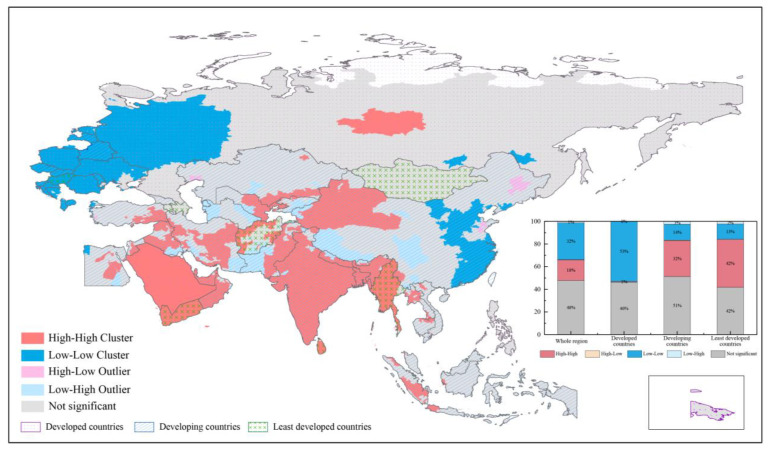
The cluster map of slope for PWEL of PM_2.5_ concentration in B&R region from 2000 to 2020.

**Figure 9 ijerph-19-02852-f009:**
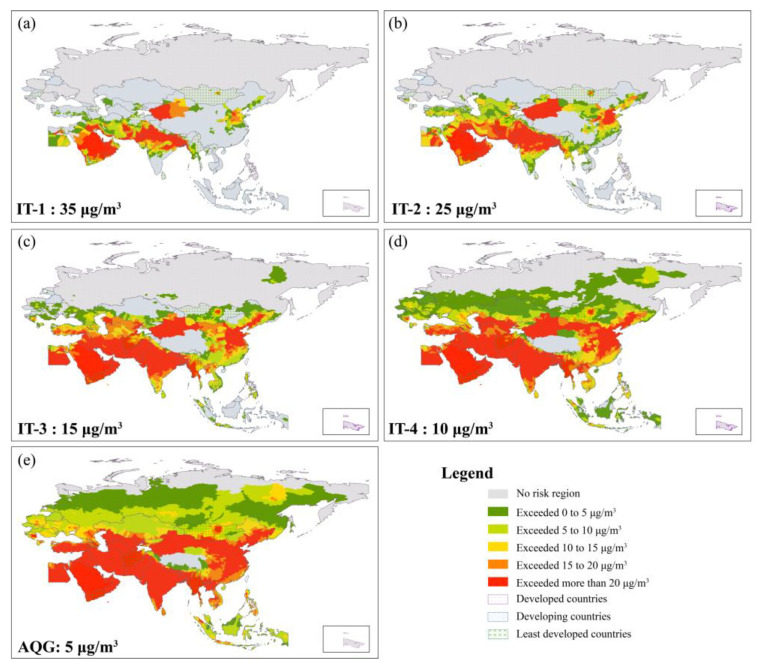
Key regions under different target of WHO Air Quality Guidelines. (**a**) IT-1 (35μg/m^3^), (**b**) IT-2 (25μg/m^3^), (**c**) IT-3 (15μg/m^3^), (**d**) IT-4 (10μg/m^3^), (**e**) AQG (5 μg/m^3^).

**Table 1 ijerph-19-02852-t001:** Recommended AQG level and interim targets of PM_2.5_ concentration.

Level	Values (μg/m^3^)
Interim target 1	35
Interim target 2	25
Interim target 3	15
Interim target 4	10
AQG level	5

## Data Availability

Data and materials used to support the findings of this study are available.

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
