# Peer review of "PM2.5 Concentration Exposure over the Belt and Road Region from 2000 to 2020"

_ijerph, 2022, doi:10.3390/ijerph19052852_

Round 1
Reviewer 1 Report
see attachment

Author Response
This study aimed to investigate the evolutionary relationships of PM2.5 pollution and its population-weighted exposure level (PWEL) over B&R based on satellite-derived PM2.5 concentration, and identify the key regions for exposure control in the future. The authors did very valuable work on collecting and presenting the material, but in the current state it still suffers from a lack of system and logic in the presentation. The result and discussion do not support the introduction, Thus, it is recommended to undertake a major revision, rewrite the paper in a manner facilitating a further use of this undeniably valuable set of information.
Author’s response: Thanks for the reviewer’s comments. According to your valuable comments, we have rewritten the result and discussion section of the manuscript, seen in the revised manuscript.
Detailed comments:
- Improve the quality English writing, too many grammatical errors.
Author’s response: Thanks for the reviewer’s comments. We have carefully revised the grammatical errors in the manuscript and also asked a native English speaker to help improve the writing quality.
- Some the analysis was inappropriate. Such as the “3.3. The evolutionary relationships based on PWEL of PM2.5 concentration” the study used the line regression to calculated the slope of PWEL during 2000 to 2020, more study had confirmed that air pollution changes was curvilinear during 2000 to 2020 (Zhang, Q., Y. Zheng, D. Tong, M. Shao, S. Wang, Y. Zhang, X. Xu, J. Wang, H. He, W. Liu, Y. Ding, Y. Lei, J. Li, Z. Wang, X. Zhang, Y. Wang, J. Cheng, Y. Liu, Q. Shi, L. Yan, G. Geng, C. Hong, M. Li, F. Liu, B. Zheng, J. Cao, A. Ding, J. Gao, Q. Fu, J. Huo, B. Liu, Z. Liu, F. Yang, K. He & J. Hao (2019) Drivers of improved PM2.5 air quality in China from 2013 to 2017. Proceedings of the National Academy of Sciences of the United States of America, 116, 24463-24469.()
Author’s response: Thanks for the reviewer’s comments. According to the reviewer’s suggestion, we carefully read the article titled “Drivers of improved PM2.5 air quality in China from 2013 to 2017”, this article mainly discussed the factors of PM2.5 concentration decreasing during the period between 2013 to 2017, and the effects of these factors to air quality improving were nonlinear. While, our study focused on exploring the long time series variations of PM2.5 concentration at a macro scale. In this case, we use the linear model to calculated the slope of PM2.5 concentration refer to a classic research article which published on the top journal in environmental field (Remote Sensing of Environment). Undoubtedly, the problem of nonlinear feature of PM2.5 concentration variation which pointed out by the reviewer also needs to be solved, in future research, this issue can be discussed separately.
Reference:
Peng, J., S. Chen, H. Lv, Y. Liu & J. Wu (2016) Spatiotemporal patterns of remotely sensed PM2.5 concentration in China from 1999 to 2011. Remote Sensing of Environment, 174, 109-121.
- Lines183-185 suggest to delete “developed countries, developing countries and least developed countries,”
Author’s response: Revised.
- Lins 245-246, suggest to delete “developed countries, developing countries and least developed countries,”
Author’s response: Revised.
Reviewer 2 Report
Suggestions are as follows:
- Abbreviations should be defined at the first which are shown in the text (e.g., PWEL, AQG).
- Line 81, on what basis are these countries divided into developed, developing and least developed groups? Please indicate the references.
- Section 2.3.4, Please clearly describe meanings of IT1-4 and their respective concentrations.
- The resolution of figures is low. Please provide hi-res pictures.
- Lines 218-228, the statement is confusing. It is supposed to talk about decrease in PM2.5, but the opposite word, "increase," is in there. Please check for the correctness.
- About the clusters, the authors talk about high-high and low-low clusters, but do not say much for high-low or low-high clusters. Do they mean something important or anything interesting?
- Are the spatial and temporal patterns of original and PWEL PM2.5 concentrations related to B&R? This paper does not seem to mention about it.
- Declaration at the end of paper needs checking and correcting. inappropriate default description should be deleted.
Author Response
- Abbreviations should be defined at the first which are shown in the text (e.g., PWEL, AQG).
Author’s response: Thanks for the reviewer’s comments, we have carefully checked the abbreviations and all abbreviations are defined in the first of text.
- Line 81, on what basis are these countries divided into developed, developing and least developed groups? Please indicate the references.
Author’s response: The developed, developing and least developed groups were classified according to the income level (classified low income countries to least developed group, upper and lower middle income countries to developing group and high income countries to developed group) based on gross domestic product (GDP), the data were collected from The Word Bank DataBank (https://www.worldbank.org/). According to reviewer’s suggestion, we also added detailed information in the revised manuscript.
- Section 2.3.4, Please clearly describe meanings of IT1-4 and their respective concentrations.
Author’s response: Thanks for the reviewer’s comments. According to the latest AQG published by WHO in September 2021, the AQG level of PM2.5 concentration was set to 5μg/m3. Meanwhile, WHO also provides four interim targets (35μg/m3, 25μg/m3, 15μg/m3, 10μg/m3) as “incremental steps in progressive reduction of air pollution”. The interim targets should be regarded as steps towards ultimately achieving AQG levels in the future, rather than as end targets. We added the above detailed descriptions of AQG levels and the meanings of the Interim targets in section 2.3.4, seen in the revised manuscript.
- The resolution of figures is low. Please provide hi-res pictures.
Author’s response: All the figures were replaced to high-resolution version.
- Lines 218-228, the statement is confusing. It is supposed to talk about decrease in PM5, but the opposite word, "increase," is in there. Please check for the correctness.
Author’s response: Revised.
- About the clusters, the authors talk about high-high and low-low clusters, but do not say much for high-low or low-high clusters. Do they mean something important or anything interesting?
Author’s response: Thanks for the reviewer’s comments, the detailed results of high-low and low-high outliers were added in the revised manuscript.
- Are the spatial and temporal patterns of original and PWEL PM5 concentrations related to B&R? This paper does not seem to mention about it.
Author’s response: Thanks for the reviewer’s comments, the discussion of relationship between original and PWEL PM2.5 concentrations written in the third paragraph of section discussion.
- Declaration at the end of paper needs checking and correcting. inappropriate default description should be deleted.
Author’s response: Revised.
Reviewer 3 Report
I found some merits in this methodology and results. In my opinion, this paper has a good potential to be published in the journal. The statistical analysis performed is just enough but I find the application in this field original. However. I have also some concerns about several parts of the manuscript. In my opinion, the manuscript structure is good, but you must improve the environmental aspect of air quality in urban areas. in the introduction, it is necessary to have information about pollutants that get worse air quality in urban areas. Moreover, it is necessary to add some information about monitoring stations. I suggest some papers that must be added in the introduction in order to improve its content and make this work more complete:
Comparative analyses of urban air quality monitoring system: Passive sampling and continuous monitoring stations.
Air quality data for Catania: Analysis and investigation case study 2012-2013.
The inscriptions of figure 9 must be written in readable mode. If you improve the weak points, this manuscript will deserve to be published in this journal.
Author Response
I found some merits in this methodology and results. In my opinion, this paper has a good potential to be published in the journal.
The statistical analysis performed is just enough but I find the application in this field original. However, I have also some concerns about several parts of the manuscript. In my opinion, the manuscript structure is good, but you must improve the environmental aspect of air quality in urban areas. in the introduction, it is necessary to have information about pollutants that get worse air quality in urban areas. Moreover, it is necessary to add some information about monitoring stations.
Author’s response: Thanks for the reviewer’s valuable comments. According to reviewer’s suggestion, we added some information that the reviewer mentioned above in the revised manuscript.
I suggest some papers that must be added in the introduction in order to improve its content and make this work more complete:
Comparative analyses of urban air quality monitoring system: Passive sampling and continuous monitoring stations.
Air quality data for Catania: Analysis and investigation case study 2012-2013.
Author’s response: Above references were added in the section of introduction.
The inscriptions of figure 9 must be written in readable mode. If you improve the weak points, this manuscript will deserve to be published in this journal.
Author’s response: Thanks for the reviewer’s valuable comments. We have rewritten the inscriptions of figure 9 according to reviewer’s suggestion.
Reviewer 4 Report
This is a study that is outdated and in 2021 the WHO established new Guideline values.
https://www.who.int/es/news-room/fact-sheets/detail/ambient-(outdoor)-air-quality-and-health
that the authors do not contemplate in this 2022 study.
Specifically, for PM2.5 the annual value established is 5 micg/m3 and not 10 micg/m3 as stated in the manuscript.
The authors should redo their study with these new guideline values.
Author Response
This is a study that is outdated and in 2021 the WHO established new Guideline values.
https://www.who.int/es/news-room/fact-sheets/detail/ambient-(outdoor)-air-quality-and-health
that the authors do not contemplate in this 2022 study.
Specifically, for PM2.5 the annual value established is 5 micg/m3 and not 10 micg/m3 as stated in the manuscript.
The authors should redo their study with these new guideline values.
Author’s response: Thanks for the reviewer’s valuable comments. According to the reviewer’s suggestion, we carefully studied the latest document of WHO AQG. Then, we have added the results of establishing the AQG as 5μg/m3, and revised related results in the new version of manuscript.
Round 2
Reviewer 4 Report
The authors have redone the results according to the new WHO guideline values.
The manuscript can be accepted for publication.